
# No Robust Evidence of Future Changes in Major Stratospheric Sudden Warmings: A Multi-model Assessment from CCMI

Blanca Ayarzagüena[1,2a], Lorenzo M. Polvani[3], Ulrike Langematz[4], Hideharu Akiyoshi[5], Slimane Bekki[6], Neal Butchart[7], Martin Dameris[8] Makoto Deushi[9], Steven C. Hardiman[7], Patrick Jöckel[8], Andrew Klekociuk[10], Marion Marchand[6], Martine Michou[11], Olaf Morgenstern[12], Fiona M. O'Connor[7], Luke D. Oman[13], David A. Plummer[14], Laura Revell[15,16], Eugene Rozanov[15], David Saint-Martin[11], John Scinocca[14], Andrea Stenke[15], Kane Stone[17,18b], Yousuke Yamashita[5c], Kohei Yoshida[9] and Guang Zeng[12]

[1] Dpto. Física de la Tierra y Astrofísica, Universidad Complutense de Madrid, Madrid, Spain.
[2] Instituto de Geociencias (IGEO), CSIC-UCM, Madrid, Spain.
[3] Columbia University, New York, USA.
[4] Institut für Meteorologie, Freie Universität Berlin, Berlin, Germany.
[5] National Institute for Environmental Studies (NIES), Tsukuba, Japan.
[6] LATMOS, Institut Pierre Simon Laplace (IPSL), Paris, France.
[7] Met Office Hadley Centre (MOHC), Exeter, UK.
[8] Institut für Physik der Atmosphäre, Deutsches Zentrum für Luft- und Raumfahrt (DLR), Oberpfaffenhofen, Germany.
[9] Meteorological Research Institute (MRI), Tsukuba, Japan.
[10] Australian Antarctic Division, Kingston Tasmania, Australia.
[11] CNRM UMR 3589, Météo-France/CNRS, Toulouse, France.
[12] National Institute of Water and Atmospheric Research (NIWA), Wellington, New Zealand.
[13] National Aeronautics and Space Administration Goddard Space Flight Center (NASA GSFC), Greenbelt, Maryland, USA.
[14] Environment and Climate Change Canada, Montréal, Canada.
[15] Institute for Atmospheric and Climate Science, ETH Zürich (ETHZ), Switzerland.
[16] Bodeker Scientific, Christchurch, New Zealand
[17] School of Earth Sciences, University of Melbourne, Melbourne, Australia
[18] ARC Centre of Excellence for Climate System Science, Sydney, Australia

[a] previously at: College of Engineering, Mathematics and Physical Sciences, University of Exeter, Exeter, United Kingdom.
[b] now at: Massachusetts Institute of Technology (MIT), Boston, Massachusetts, USA
[c] now at: Japan Agency for Marine-Earth Science and Technology (JAMSTEC), Yokohama, Japan

*Correspondence to*: Blanca Ayarzagüena (bayarzag@ucm.es)

**Abstract.**

Major mid-winter stratospheric sudden warmings (SSWs) are the largest instance of wintertime variability in the Arctic stratosphere. Because SSWs are able to cause significant surface weather anomalies on intra-seasonal time scales, several previous studies have focused on their potential future change, as might be induced by anthropogenic forcings. However, a wide range of results have been reported, from a future increase in the frequency of SSWs to an actual decrease. Several factors might explain these contradictory results, notably the use of different metrics for the identification of SSWs, and the impact of large climatological biases in single-model studies. To bring some clarity, we here revisit the question of future SSWs changes, using an identical set of metrics applied consistently across 12 different models participating in the



Chemistry Climate Model Initiative. Our analysis reveals that no statistically significant change in the frequency of SSWs will occur over the 21$^{st}$ century, irrespective of the metric used for the identification of the event. Changes in other SSWs characteristics, such as their duration and the tropospheric forcing, are also assessed: again, we find no evidence of future changes over the 21$^{st}$ century.

# 1 Introduction

Stratospheric sudden warmings (SSWs) are the largest manifestation of the internal variability of the wintertime polar stratosphere in the Northern Hemisphere, consisting of a very rapid temperature increase accompanied by a reversal of the westerly wintertime circulation (the polar vortex). In observations, SSWs occur roughly with a frequency of 6 SSWs per decade (e.g., Charlton and Polvani, 2007). However, large variability on intra- and inter-decadal time scales has been reported (Labitzke and Naujokat, 2000; Schimanke et al., 2011).

SSWs also play an important role in the dynamical coupling between the stratosphere and troposphere. They are known to originate from precursors in the troposphere, as SSWs are triggered by an anomalously high injection of tropospheric waves that propagate into the stratosphere where they deposit momentum and energy, decelerating the mean flow (Matsuno, 1971; Polvani and Waugh, 2004). More importantly, however, their effects are not restricted to the stratosphere: SSWs also impact the tropospheric circulation and surface climate for up to two months (e.g., Baldwin and Dunkerton, 2001). Given their 55 importance for seasonal forecasting, SSWs have been studied with great interest, as they are likely to provide a source of improved weather forecasts at intraseasonal scales (Sigmond et al., 2013).

One question of particular relevance is whether SSWs will change in the future, as a consequence of increasing greenhouse gases (GHG) concentrations and ozone recovery. The answer to this question has proven elusive since the first studies over two decades ago. While Mahfouf et al. (1994) found an increase in the frequency of SSWs under doubled $CO_2$
conditions, Rind et al. (1998) reported a decrease, and Butchart et al. (2000) did not find any change that might be attributed to increasing GHG concentrations. And, in spite of an improved stratospheric representation and more realistic model features in the last decade, a clear consensus as to future SSW changes is still missing (Charlton-Perez et al., 2008; Bell et al., 2010; SPARC CCMVal, 2010; Mitchell et al. 2012a and b; Hansen et al., 2014).

Several potential reasons that might explain the disparity in the projected SSW changes have been proposed in the 65 literature. One is the combination of different aspects of future climate change with opposing effects on the Arctic stratosphere, such as the projected ozone recovery, increasing GHG concentrations and their induced changes in global sea surface temperatures. These result in a weak polar stratospheric response to climate change (Mitchell et al., 2012a, Ayarzagüena et al., 2013). Consequently, individual models yield different future projections of SSW changes, depending on the relative importance of these competing effects in each model. Hence, any result obtained with a single model needs to be taken with 70 much caution.



Another potential explanation for the discrepancies stems from the criterion chosen for the identification of SSWs. As shown in Butler et al. (2015), the identification of SSWs can be sensitive to the method used. It was found to depend on the meteorological variable chosen for analysis, and also on whether the identification criterion entails total fields and a fixed threshold (absolute criterion), or anomalies relative to a changing climatology (relative criterion). For instance, the traditional

criterion of the World Meteorological Criterion (hereafter WMO criterion, McInturff, 1978) requires the reversal of both zonal-mean zonal wind at 60ºN and 10hPa and the meridional gradient of zonal mean temperature between 60ºN and the pole at the same level. This criterion was empirically developed from the observations in the last several decades, and was applied in historical stratospheric analyses (e.g., Labitzke, 1981). Recent studies have continued using the WMO criterion although many of them have only imposed the reversal of the wind for the SSW identification (e.g., Charlton and Polvani, 2007). Because of

its simplicity and its dynamical insight, the WMO criterion (and its recent simplified version) is the most commonly used criterion in modelling studies as well. However, such an absolute metric might not always be the best choice to measure the polar stratospheric variability in these studies, as it does not account for potential model biases in the polar vortex climatology, or possible changes in this climatology in the future projections (McLandress and Shepherd, 2009; Mitchell et al., 2012a; Butler et al., 2015). An analysis with the Canadian Middle Atmosphere Model by McLandress and Shepherd (2009) showed

that the frequency of SSWs may or may not change depending on the detection index.

The purpose of this study, therefore, is to revisit the question of possible future SSW changes, taking these issues into consideration. Seeking a robust answer, we employ three different SSW identification criteria (both absolute and relative) and apply them consistently to the output from 12 state-of-the-art climate models (contributing to the Chemistry Climate Model Initiative, CCMI). Interactive stratospheric chemistry, which is present in all the CCMI models, makes them the most realistic

in terms of stratospheric processes. In addition the CCMI models are improved compared to their counterparts which participated in the previous Chemistry Climate Model Validation-2 programme (CCMVal-2). In particular, several CCMI models are coupled to interactive ocean modules, and the vertical resolution of many models has been increased (Morgenstern et al. 2017). The structure of the paper is as follows: In Section 2 the data and methodology used in the analysis are described. The main results are shown in Section 3, and Section 4 includes the discussion and the most important conclusions derived

from the analysis.

## 2 Data and methodology

### 2.1. Data description

Our study is based on the analysis of the transient REF-C2 simulation of 12 CCMI models (cf. Table 1; for more details see Morgenstern et al., 2017). The REF-C2 runs extend from 1960 to 2099 or 2100 for most models (except for the

IPSL-LMDZ-REPROBUS model that terminates the run in 2095), and include natural and anthropogenic forcings following the CCMI specifications (Eyring et al., 2013). In particular, GHG concentrations and surface mixing ratios of ozone depleting substances (ODS) are based on observations until 2000, and on the Representative Concentration Pathway 6.0 (RCP6.0,



Meinshausen et al., 2011) and A1 (WMO, 2011) scenarios, respectively, from 2000 to 2100. Solar variability is included in most of the models. Depending on the characteristics and performance of the models, sea surface temperatures (SSTs) and the

quasi-biennial oscillation (QBO) are prescribed or internally generated. Future changes in frequency and other features of SSWs are obtained by comparing the last 40 winters of each run (denoted as "the future") to the first 40 winters (denoted as "the past"). Unless otherwise stated, anomalies are calculated from the climatology of the corresponding 40-year period. A Student's t-test is applied to determine if the future changes are statistically significant in all cases except for the duration of SSWs where we applied a Wilcoxon ranked-sum test. The performance of the models in reproducing SSWs characteristics for

the past period (1960-2000) is assessed by comparing the models to the ERA-40 and JRA-55 reanalyses (Uppala et al., 2005; Kobayashi et al., 2015). Both reanalyses extend back of 1979, covering the past period of our study. Among the few reanalyses that have available data in the pre-satellite era, ERA-40 and JRA-55 are the most suitable for middle atmosphere analyses because they have a higher top level and vertical resolution (Fujiwara et al., 2017).

## 2.2 Criteria for the detection of SSWs

As the detection of SSWs is somewhat sensitive to the chosen criterion, we use three different criteria to ensure that the conclusions regarding future changes are the same irrespective of the metric. The criteria we use are described in Butler et al. (2015) and as follows.

*1) WMO (World Meteorological Organization) criterion*

SSWs are identified when the zonal-mean zonal wind at 10 hPa and 60°N and the zonal-mean temperature difference

between 60°N and the pole at the same level reverse. Two events must be separated by at least 20 consecutive days of westerly winds. Only events from November to March are considered. Stratospheric final warmings are excluded by imposing at least 10 days with westerly winds after the occurrence of a SSW and before 30 April, to ensure the recovery of the polar vortex before its final breakup. The onset date of the event corresponds to the first day of the wind reversal.

*2) Polar cap zonal wind reversal (u6090N)*

SSWs are identified when the zonal wind at 10 hPa averaged over the polar cap (60°N-90°N) reverses. The separation of events and the exclusion of stratospheric final warmings are done in the same way as for the WMO criterion.

*3) Polar cap 10hPa geopotential (ZPOL)*

SSWs are identified based on the polar cap standardized anomalies of 10 hPa geopotential height. The anomalies are detrended and computed following Gerber et al. (2010). A SSW is detected if the anomalies exceed three standard deviations

of the climatological January to March geopotential height (Thompson et al. 2002).

Note that WMO and u6090N are absolute SSWs criteria, whereas ZPOL is a relative SSW one.

## 2.3 Other SSW characteristics

Beyond their frequency, we also study if the other key characteristics of SSWs, such as duration and tropospheric forcing, will change in the future. The considered events in all features are those identified by the WMO criterion, because it



is a popular criterion and, as will be shown later, the conclusions relative to the frequency results are not different from those obtained for the other two criteria. These are the metrics/diagnostics applied:

    *1. Duration:*

The duration of the events is computed by the number of consecutive days of easterly wind regime at 60°N and 10 hPa as in Charlton et al. (2007).

*2. Tropospheric forcing*

The analysis of the tropospheric forcing is based on the evolution of the anomalous eddy heat flux at 100 hPa averaged between 45° and 75°N (aHF100) before and after the occurrence of SSWs. aHF100 is a measure of the injection of tropospheric wave activity into the stratosphere (Hu and Tung, 2003).

## 3. Future changes in the main characteristics of SSWs

**3.1 Mean frequency**

      We start by considering the frequency of SSWs, and whether it is projected to change as a consequence of anthropogenic forcings. For this purpose, we have identified SSWs in the 12 models listed in Table 1, for the past and future periods, according to the three criteria presented in Section 2.2. Figure 1 shows the mean frequency of SSWs for each case.

      In spite of some differences among the criteria, there appears to be a suggestion of a small increase in frequency in
the multimodel mean (hereafter MM), but this tendency is not statistically significant at the 95% confidence level for any of the criteria, either absolute or relative. Also, while most models show a small increase in the frequency of SSWs in the future (10 of 12 models for the WMO criterion; 9 of 12 in the u6090N criterion; and 7 of 12 for the ZPOL), most of those changes are not statistically significant. Specifically, none of the models displays a statistically significant future change for the relative criterion (ZPOL) (Fig. 1c), only 3 out of 12 models show a significant increase for the WMO criterion (NIWA-UKCA, EMAC-
L90 and CMAM) (Fig. 1a), and only 2 out of 12 models for the u6090N criterion (SOCOL3, EMAC-L90) (Fig. 1b). It is, however, important to note that the NIWA-UKCA and CMAM models do not simulate a realistic frequency of SSWs when compared to reanalyses for the current climate, so they may not be a reliable indicator of possible future changes. Additionally, none of the four models (NIWA-UKCA, SOCOL3, EMAC-L90 and CMAM) shows an increase in SSWs for the three criteria simultaneously, indicating the lack of consistency for those models across the different methods. This confirms the absence of
a robust future signal regarding changes in the frequency of SSWs.

      A further comparison of the results for the different criteria for the past period confirms the findings of previous studies (e.g. McLandress and Shepherd, 2009) which showed that models' biases in mean state and variability affect the frequency values for the absolute criteria, since the different models show a wide range of SSW frequency values in the past period (see Fig. S1). For instance, CCSRNIES-MIROC3.2 and NIWA-UKCA show very low SSW frequencies in agreement
with the fact that the polar vortex in these models is much stronger than in the reanalyses, and the opposite is seen for ACCESS CCM, CMAM and CNRM-CCM (Fig S2). Note the good agreement between the JRA-55 and ERA-40 reanalyses. Conversely,




SSW frequencies computed with the relative ZPOL criterion are more similar across the models, as they are less affected by climatological model biases. Interestingly, note how the values for the relative criterion are somewhat lower in models than in the reanalyses. Since the threshold for selecting events is based on the latter, this suggests that models may be underestimating the variability of the Arctic polar stratosphere.


Finally, it is worth highlighting that nearly identical results to the ones obtained with the WMO criterion are found, for both past and future periods, when only the reversal of the wind at 60°N and 10hPa (Charlton and Polvani, 2007) is used as the identification criterion. It is reassuring to report that the additional temperature constraint imposed in the WMO criterion does not significantly alter the frequency of SSWs, even for the future climates. This means that most recent studies, which have used the simpler method and considered the reversal of the wind as the sole quantity for identifying SSWs, would have likely reached the same conclusions had they used the more precise WMO criterion, and can thus be considered valid.


## 3.2 Duration

Next, we turn to the duration of SSWs, for which the results are shown in Fig. 2, for the past and future. In each period, we notice a considerable spread across the models; nonetheless, the MM value for the past period falls within the interval of reanalyses values ±1.5 standard error. Note, however, the variability within each model is larger than that across the models. This is particularly true for the NIWA-UKCA and CCSRNIES-MIROC3.2 models, possibly as a consequence of the low number of SSWs simulated by these two models. MRI-ESM1r1 also shows a large variability in SSW duration, but only in the past period.


The key message from Fig. 2 is that the duration of SSWs does not change in the future, using the canonical 95% confidence level. Nevertheless, as in the case of the mean frequency, more than half of the models (7 out of 12) agree on the sign of the future change in the SSW duration (they indicate that it will be slightly shorter), but this change in the MM is not statistically significant at the 95% confidence level.


## 3.3 Tropospheric forcing

Since SSWs are usually triggered by anomalously high tropospheric wave activity entering the stratosphere in the weeks preceding the events (Matsuno, 1971; Polvani and Waugh 2004), we have analyzed the possible future changes in the injection of wave activity (aHF100) in the course of the occurrence of these events for the MM (Fig.3). The results do not show a statistically significant change in any aspect of the anomalous wave activity preceding SSWs in the MM and in the individual models (not shown). In particular, neither the strong peak of aHF100 of the MM in the 10 days prior to the occurrence of events nor the general time evolution of the aHF100 are projected to change in the future (Fig.3a). Hence the common, but not statistically significant, trend of models towards shorter future SSWs mentioned above cannot be explained by changes in tropospheric forcing. Additionally, when examining the two first zonal wavenumber components of the anomalous HF100, no significant future changes are found either (Fig.3b).





Model projections of future aHF100 are reliable because models are able to simulate the tropospheric forcing of these events reasonably well (Fig.3). Only a few discrepancies can be seen between the MM and the mean of JRA-55 and ERA-40

reanalyses (Reanalyses Mean, RM, black curve). Note that we include the average of JRA-55 and ERA-40 because they show very similar results and we avoid confusion by including too many lines in the same plot. One of the discrepancies between MM and RM is that the strong peak in aHF100 in the 5 days prior to the occurrence of SSWs is weaker in the models than in observations. The reanalyses also show a secondary peak of aHF100 between -20 and -10 days that does not appear in the MM. Additionally, the contribution of the wavenumber 1 (WN1) component to the strongest wave pulse is similar or even

stronger than in the reanalyses (Fig.3b), but the wavenumber 2 (WN2) in the models is much weaker than in the RM. This explains the weaker total value of aHF100 in the MM than in the RM. Nevertheless, the RM is only one realization averaged over 40 years and the MM corresponds to the average over many more realizations. Thus, the multi-model/individual realization spread possibly account for at least partially these two mismatches between MM and RM. In any case, the models show no changes between the past and the future.

## 4. Discussion and conclusions

We have revisited the question of whether SSWs will change in the future, analysing 12 state-of-the-art stratosphere resolving models that participated in CCMI. To obtain robust results, we have used three different identification criteria (two absolute and one relative) and have applied them consistently across all 12 models. In summary, our analysis reveals that:

- No statistically significant changes in the frequency of occurrence of SSWs are to be expected in the coming decades
and until the end of the 21 century. This result is robust, as it is obtained with three different identification criteria.
- Other features of SSWs, such as their duration and the tropospheric precursor wave fluxes, do not change in the future either in the model simulations, in agreement with other studies such as McLandress and Shepherd (2009) or Bell et al. (2010).

Despite the lack of statistical significant changes in the frequency of SSWs, both the MM and the majority of the

models analysed show a slight increase in frequency across all criteria. A similar result was reported by Kim et al. (2017), who analysed the change in SSW frequency in some CMIP5 models by identifying the events based either on the reversal of the wind or the vortex deceleration. Looking at changes in the daily climatology of the zonal mean zonal wind at 10 hPa (Figs. 4a and S3), the MM and individual model simulations also provide a consistent picture, with a robust weakening of the polar night jet (PNJ) from mid-December until mid-March, the deceleration being particularly strong between mid-December and

mid-February; this is in agreement with previous CMIP5 results (Manzini et al., 2014). This deceleration is, however, only statistically significant in less than half of the models (Fig. S3), explaining why we do not find a significant change in the tropospheric forcing of SSWs (Fig. 3). To determine whether these changes in the climatology of wintertime PNJ might be associated with changes in SSWs frequency, the future-minus-past difference plots of the climatological wind are shown separately for winters with and without SSWs (Fig. 4b and c, respectively). We find a weakening of the PNJ in midwinter in



both cases: this allows us to conclude that the future deceleration of the PNJ is not a consequence of a higher frequency of SSWs. This deceleration might be related to a general increase in the total stratospheric variability that, in the case of winters without SSWs, would correspond to a higher frequency of minor warmings. However, this possibility is unlikely because we do not find a robust future increase in the standard deviation of zonal-mean zonal wind at 10hPa across the models (not shown). Perhaps the future deceleration of the PNJ might explain the statistically significant increase in SSWs in a few models, using

the absolute criteria. In any case, these signals are small and it is nearly impossible to untangle the cause and the effect, as these changes occur simultaneously.

More importantly our findings dispel, to a large degree, the confusion in the literature regarding future SSW changes, and suggest that previous reports of significant changes are likely to be artefacts, caused by biases associated with individual models, or by flaws in the identification methods used (or both). Note that although the key finding of our study – i.e. that

anthropogenic forcings will not affect SSWs over the 21st century – is a null result, it is by no means uninteresting. Just to offer one example: Kang and Tziperman (2017) have recently proposed that future changes in the Madden Julian Oscillation (which are expected to occur with increased levels of $CO_2$ in the atmosphere) will cause an increased occurrence of SSWs. While their conclusion may be correct, our findings indicate that it can be misleading to project changes in the SSWs on the basis of a single mechanism: the complexity of the climate system is such that multiple mechanisms may be at play, with likely opposite

effects which may result in net changes that are not statistically significant.

One may argue that the lack of a statistically significant future change in our study could be explained, at least partially, by the high interannual variability of the boreal polar stratosphere in 40-year periods (e.g., Langematz and Kunze, 2006), or perhaps by the natural variability on longer time-scales coming from other subcomponents of the climate system (e.g.: Schimanke et al., 2011). As shown in a recent paper, 10 identically forced model simulations, over the 50-year period

1952-2003, exhibit great differences in the number of SSWs, and these differences are solely due to internal variability (Polvani et al, 2017). This means that the 40 years of observations at our disposal may not represent the mean of a distribution, but could happen to be an outlier. Needless to say, we have no means of determining whether this is the case, as we do not have long enough observations.

One might also object that the forcing in the scenario used of our runs (RCP6.0) is not extreme enough to produce a

significant signal in the frequency and duration of SSWs, but that significant change would occur with stronger forcing. For instance, one may think that this signal might become significant under the RCP8.5 scenario. Although we cannot rule out this possibility, it seems improbable based on a similar lack of significance in the results documented for that very extreme scenario by several previous studies (Mitchell et al. 2012a; Ayarzagüena et al. 2013, Hansen et al. 2014; Kim et al. 2017). Nevertheless, it would be hard to verify the hypothesis because of the low number of CCMI RCP8.5 simulations available.

Finally, in the last years much activity has been devoted to search for novel criteria for the identification of SSWs (Butler et al., 2015). One of the reasons given to justify the implementation of a new metrics was that the traditional WMO criterion was not appropriate for modelling studies, as it was based on observationally chosen parameters, such as the location of the polar night jet. However, our results show that this criterion performs well under a changing climate, provided models



are able to reproduce correctly the past stratospheric variability. Thus, considering the good agreement among the three criteria used here on the lack of change in future SSWs, and given the dynamical implications for the propagation of planetary waves into the stratosphere, we suggest that the WMO criterion is appropriate for the study of SSWs in the future if the model can represent well the stratospheric variability. Furthermore, since the simplest (and most commonly used) criterion, involving only the zonal winds (Charlton and Polvani, 2007), yields identical results as the WMO criterion, one could argue that the simplest method may suffice in most cases for the study of SSWs, and that more complex criteria might not be worth the

trouble. A similar conclusion was reached, independently, by Butler and Gerber (2018) who methodically assessed different metrics and concluded that the simplest algorithm is within the optimal range.

**Data availability.** Data of this manuscript has been mostly downloaded from the Centre for Environmental Data Analysis (CEDA, 2017; http://catalogue.ceda.ac.uk/uuid/9cc6b94df0f4469d8066d69b5df879d5 ) or supplied directly by the co-authors. For instructions for access to this archive see http://blogs.reading.ac.uk/ccmi/badc-data-access. The data supplied by the co-

authors will in due course be uploaded to the CEDA archive.

**Acknowledgements**

We acknowledge the modeling groups for making their simulations available for this analysis, the joint WCRP SPARC/IGAC Chemistry-Climate Model Initiative (CCMI) for organizing and coordinating the model data analysis activity, and the British Atmospheric Data Centre (BADC) for collecting and archiving the CCMI model output. BA was funded by the European

Project 603557-STRATOCLIM under the FP7-ENV.2013.6.1-2 programme and "Ayudas para la contratación de personal postdoctoral en formación en docencia e investigación en departamentos de la Universidad Complutense de Madrid". LMP is grateful for the continued support of the US National Science Foundation. The work of NB, SCH, and FOC was supported by the Joint BEIS/Defra Met Office Hadley Centre Climate Programme (GA01101). NB and SCH were supported by the European Community within the StratoClim project (Grant 603557). OM and GZ acknowledge the UK Met Office for use of

the MetUM. This research was supported by the NZ Government's Strategic Science Investment Fund (SSIF) through the NIWA programme CACV. OM acknowledges funding by the New Zealand Royal Society Marsden Fund (grant 12-NIW-006) and by the Deep South National Science Challenge (http://www.deepsouthchallenge.co.nz). The authors wish to acknowledge the contribution of NeSI high-performance computing facilities to the results of this research. New Zealand's national facilities are provided by the New Zealand eScience Infrastructure (NeSI) and funded jointly by NeSI's collaborator institutions and

through the Ministry of Business, Innovation & Employment's Research Infrastructure programme (https://www.nesi.org.nz). The EMAC simulations have been performed at the German Climate Computing Centre (DKRZ) through support from the Bundesministerium für Bildung und Forschung (BMBF). DKRZ and its scientific steering committee are gratefully acknowledged for providing the HPC and data archiving resources for the consortial project ESCiMo (Earth System Chemistry integrated Modelling). CCSRNIES's research was supported by the Environment Research and Technology Development



Funds of the Ministry of the Environment (2-1303) and Environment Restoration and Conservation Agency (2-1709), Japan, and computations were performed on NEC-SX9/A(ECO) and NEC SX-ACE computers at the CGER, NIES.

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



**Tables**

Table 1. Main characteristics relative to the models and their REF-C2 simulations used in this study.

| CCMI models | Model resolution | QBO | Solar variability | SSTs |
|---|---|---|---|---|
| GEOS-CCM | 2.5° x 2°, L72 (top:0.01hPa) | Internally generated | No | Prescribed (CESM1) |
| CNRM-CCM | T42L60 (top: 0.07 hPa) | Internally generated | Yes | Prescribed (CNRM) |
| NIWA-UKCA | 3.75° x 2.5°, L60 (top: 84 km) | Internally generated | No | Coupled to ocean model |
| CCSRNIES-MIROC 3.2 | T42L34 (top: 0.012 hPa) | Nudged | Yes | Prescribed (MIROC 3.2) |
| IPSL-LMDZ-REPROBUS | 3.75° x 2.5°, L39 (top: 70 km) | Nudged | Yes | Prescribed (SRES A1b IPSL) |
| ACCESS CCM | 3.75° x 2.5°, L60 (top: 84 km) | Internally generated | No | Prescribed (HadGEM-ES2) |
| HadGEM3-ES | 1.875°x1.25°, L85 (top: 85 km) | Internally generated | Yes | Coupled to ocean model |
| SOCOL3 | T42L39 (top: 0.01hPa) | Nudged | Yes | Prescribed (CESM1(CAM5)) |
| MRI-ESM1r1 | TL159L80 (top: 0.01hPa) | Internally generated | Yes | Coupled to ocean model |
| EMAC-L47 | T42L47 (top: 0.01hPa) | Nudged | Yes | Prescribed (HadGEM2-ES) |
| EMAC-L90 | T42L90 (top: 0.01hPa) | Internally generated (slightly nudged) | Yes | Prescribed (HadGEM2-ES) |
| CMAM | T47L71 (top: 0.0575 hPa) | No | No | Prescribed (CanCM4) |



**Figures**

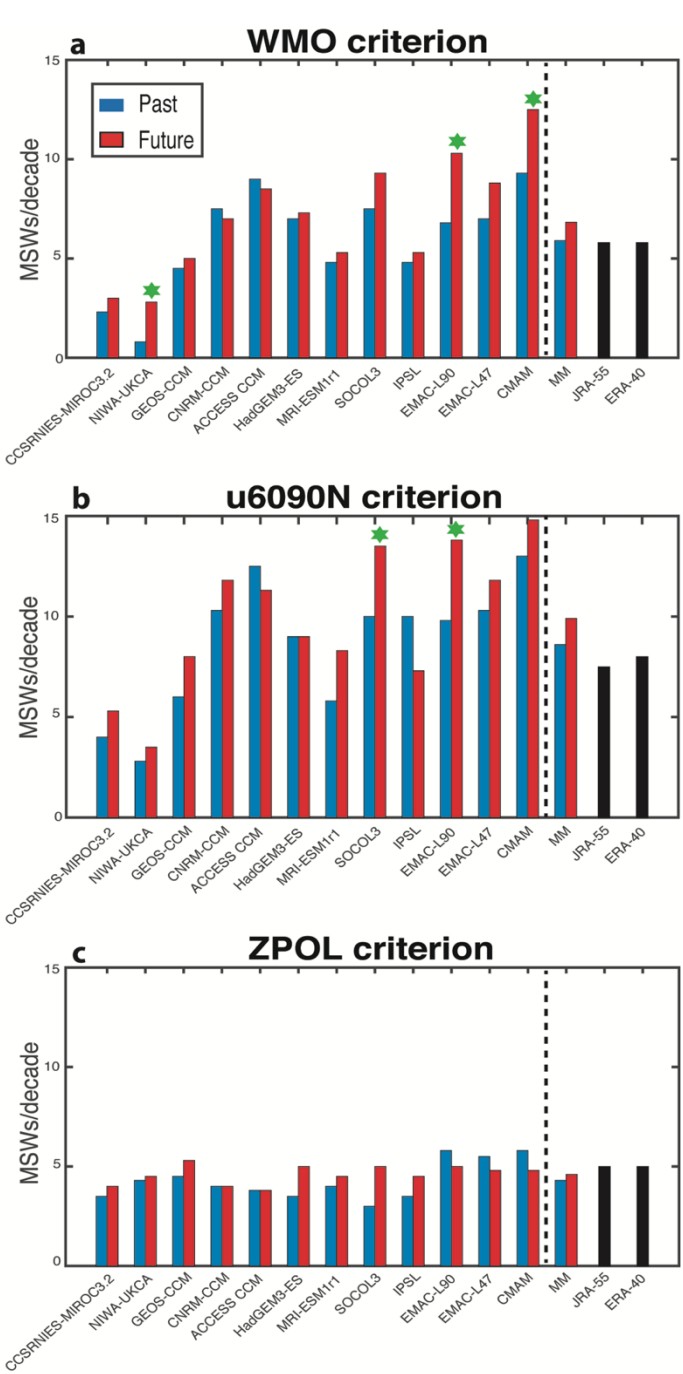


**Figure 1: (a) Mean frequency of major stratospheric warmings per decade for the past (blue bars) and the future (red bars) for all models, the multimodel mean (MM) and JRA-55 and ERA-40 reanalyses (black bars) according to the WMO criterion. (b) – (c) Same as (a) but for the u6090N and ZPOL, respectively. Green stars on top of the future bar denote a statistically significant change in the frequency of SSWs in the future at the 95% confidence level.**



**Figure 2. Duration of SSWs (in days) in each model for both periods of study. Bars denote ±1.5 standard error and green stars would indicate future values that are statistically significantly different from the past ones at the 95% confidence level (but they are absent as there are not statistically significant changes).**





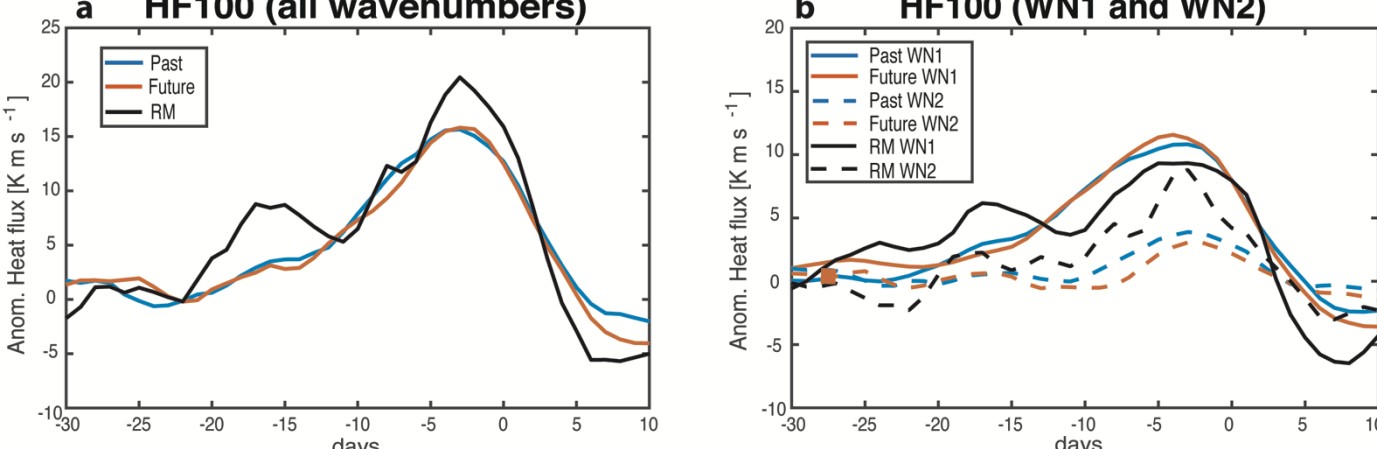

**Figure 3. (a) Multimodel mean of anomalous heat flux (K m s$^{-1}$) at 100hPa averaged over 45°N-75°N from 30 days before until 10 days after the occurrence of SSWs. (b) Same as (a) but for WN1 (solid lines) and WN2 (dashed lines) wave components. Thick lines denote statistically significant future values different from the past ones at the 95% confidence level. RM stands for the Reanalyses (JRA-55 and ERA-40) Mean.**





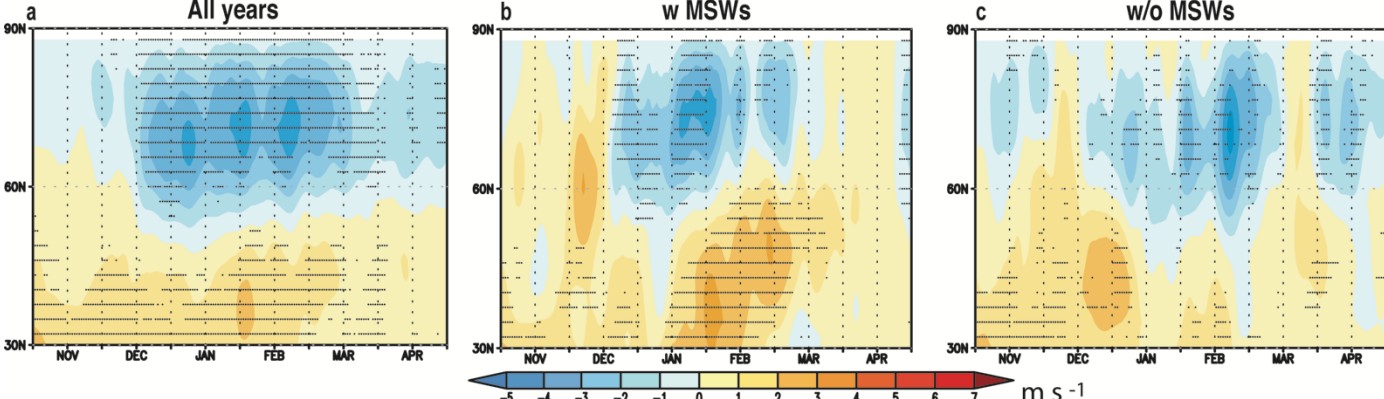

**Figure 4. (a) Multimodel mean of future-minus-past differences in the daily climatology of 5-day running mean of zonal mean zonal**
**410  wind at 10 hPa. (b) Same as (a) but only for winters with SSWs. (c) Same as (a) but for winters without SSWs. Shading interval: 1**
**m s⁻¹. Dots indicate where at least 75% of the models coincide in sign with the multimodel mean.**