# Peer review of "No Robust Evidence of Future Changes in Major Stratospheric Sudden Warmings: A Multi-model Assessment from CCM1"

_Atmospheric Chemistry and Physics, 2018_

## Referee Comment (RC1) · Anonymous Referee #1 · 21 Apr 2018

Dear Editor,

I have attached a pdf of my comments to this submission as well as included them below. Overall I think the manuscript is of value to the community and is well-written and presented. Hence, my suggestion is only of minor corrections.

General Comments This paper examines the future increase/decrease of SSWs due to anthropogenic forcing in a large ensemble of models (12) from the CCMI database. Previous studies have had opposing views on this with some finding an increase in the number of SSWs whilst others have found a decrease. To check the robustness, they use three different commonly-used SSW criteria and conclude that they do not find a

significant increase in the future number of SSWs over the course of the 21st century. I find the results to be interesting in that, by using a much larger number of models than previously used, the overall number of SSWs will remain approximately the same. The paper is well-written and the topic is well discussed, hence I suggest only minor corrections which I list below.

Specific Comments Lines51-52; not just due to anomalously large injections of wave activity from the troposphere, but can also be due to the resonance of wave activity such as that first described in Plumb (1981) and shown by Esler and Scott (2005). I would either cite this second mechanism, or just change the wording in your current sentence to be less definite. Line 53-54; Small point, but not all SSWs can impact the troposphere (e.g., Gerber et al. 2010, GRL and Hitchcock and Simpson 2014). I would instead just add the word 'can' into the sentence to change it from meaning all as it is currently: 'SSWs can also impact the...' Line 151; which criteria are absolute and which are relative? Can you somewhere distinguish between the two – from what I can see, the U6090N and WMO are absolute and ZPOL is relative. Line 184-185; at which confidence level do the models simulate statistically significant differences between the SSW duration in the past and in the future? Can you make a comment here on the 90% level? Perhaps just say how many models would become significant if this level was used. My guess is that the HADGEM3-ES and MRIESM-1r1 would be significant at close-to the 95% level as the error bars are nearly separated. Line 191; the eddy heat flux (v'T') is what you plot right? This is a proxy for the injection of wave activity. Can you make this clearer? Further, I gather from figure 3 that you use 100hPa (I think this should be included here in the text also), but given the recent paper by de la Camara et al. (2017) and Birner and Albers (2017), who suggest that 100hPa is not an ideal surface to use as it is already in the bottom of the vortex, how sensitive are you results to the level of choice? Do you get more significant results if you use a slightly lower level? Section 2.2 and Line 197; I would be interested to see what the changes between the number of splits and displacements are between the past and the future. From figure 3 it appears that the wave 1 and wave 2 forcing through 100hPa doesn't

change too much between each period and so the number of splits and displacements may not change too much either. But given the relatively short length of the paper as it is, this would be an interesting addition.

Technical Comments/Grammatical Errors Line 56; on → at Line 82; The recent paper by Kim et al. (2017) which you cite later may be a good citation here. Line 111; Could you clarify this sentence to say whether both reanalyses extend back to 1979, or that both extend back to before 1979 (and if the latter, then which year: 1960?) Line 125; Is the Polar Cap area weighted? I think it should be and it would be good to include a sentence here saying so. Line 191; 'in the course' → 'during the course'. Also, aHF100 in the text should correspond to the figure title of HF100 Line 193; None of the individual models show significantly different results? Line 209; 'no statistically significant changes' Line 220; Can you give references to the figure which shows this? Figure 1? Line 221; 'across' → 'using' Line 260; 'in the last years' → 'in recent years' Line 261; 'metrics' → 'metric'

Please also note the supplement to this comment:
https://www.atmos-chem-phys-discuss.net/acp-2018-296/acp-2018-296-RC1-supplement.pdf

———————————————

---

## Referee Comment (RC2) · Anonymous Referee #2 · 27 Apr 2018

Summary The authors look at 12 CCMi models to determine whether the frequency of sudden stratospheric warmings (SSWs) will change by the end of this century for a moderate climate change scenario. They also consider three different SSW metrics. The authors determine that there are no robust changes in the frequency of SSWs by the latter half of this century.

The paper is easy to read and the main message and methodology are clear. In general I feel positive about the paper and the conclusion but I do think the paper lacks much in-depth analysis above what has been done in previous work, most notably Kim et al. 2017. Some of the figures in the supplementary material might be worth including in

the main manuscript (Fig S3 for example). I suggest a major revision to address the points below.

General Comments 1) While this study looks at the CCMi models, which is a likely improvement in terms of interactive chemistry and stratospheric processes, it would be nice if this study more clearly explained how its analysis improves or expands upon those of Kim et al. 2017, which considered a large number of stratosphere-resolving (and non-stratosphere resolving) CMIP5 models and two different SSW definitions (and the more extreme RCP8.5 scenario rather than the RCP6.0 scenario used here). From what I can tell, the results were very similar in both studies (an increase in SSWs in future climate scenarios, though not a significant increase), though the message is quite different. While emphasizing the non-significance of the trend does seem important, the results are basically the same. It would be nice if this analysis had included some more in-depth analysis of the CCMi models in particular, maybe of whether models that had different characteristics (prescribed SSTs vs coupled, internal QBO vs nudged, solar variability or no) had different changes in SSWs. The two points below also outline some areas where more analysis could be considered.

2) This study does look at common SSW definitions, but given that the changes in mean zonal winds at 60N in Figure 4 are barely significant, it would be nice to consider zonal wind reversals at a broader range of latitudes. The authors did look at the 60-90N averaged zonal winds, but if that's cosine weighted it will be dominated by winds near 60N. 60N also seems right on the node between significant weakening winds and significant strengthening winds. It would be interesting to explore (and perhaps more clearly quantify) whether those models that showed no increases (or reductions) in SSWs had significant strengthening mid-latitude winds extending further north (seems to be true considering Fig S3). This might beg the question then, if the jet itself is shifting in the future, is maintaining a definition fixed at a particular latitude like 60N the best way to detect changes? I do agree that the polar cap anomaly/U60-90N results suggest it might not matter too much where it's defined, but exploring that sensitivity

more methodically might be useful.

3) While it does seem clear that there is no significant change in the frequency of SSWs in this analysis, the fact that there appears to be a quite robust weakening of the polar vortex (in a mean sense) is only briefly pointed out in the Discussion. It's worth keeping in mind that since the SSWs represent the tail end of the zonal wind distribution, they may be much more sensitive than the mean to small sample size and higher order moments (e.g., changes in skewness). The authors do mention that the "broadness" (variability) of the distribution does not seem to be changing over time- is it possible that the distribution becomes more skewed? I wonder though whether this weakening of the mean state could have potential climate impacts even if the most extreme events (SSWs) are not significantly changing.

Specific Comments Line 63-64: Some of the more recent papers on this topic should be cited here, including Kim et al. 2017 and Manzini et al. 2014, rather than only at the end.

Line 85: The Kim et al. 2017 results should also perhaps be mentioned here, because they investigated the definition sensitivity as well

Line 105-107: Would it be possible to consider 40-year blocks from 1960-2100—either by moving the center of the 40 years by ∼5-10 years over the full period and getting a distribution that way, or by using a smaller time period (20-30 years) and looking at the change in frequency of consecutive 30-year periods over the entire run? I wonder whether that would give you a better sense for how variable the SSW frequency can be for any given 40 year period (maybe the variability between periods is much larger than the trend between the first and last period).

Line 119-121: How did you deal with the temperature criteria here; was the zonal wind first detected and then the temperature gradient had to reverse within a certain number of days?

Line 137-140: Could any other metrics be considered in addition? These are both interesting features but other metrics like the amplitude or depth of the reversal could also be worth considering (to try to further quantify whether these events will still produce significant surface impacts in the future).

Technical Corrections Line 56: change to "weather forecasts on intraseasonal timescales" Line 111: ERA-40 and JRA-55 extend further than 1979, is that what you mean? Maybe instead of "back of", change to "beyond"? Line 208: change to "possibly accounts for at least some of the mismatch between"

---

## Author Comment (AC1) · 29 Jun 2018

**Reply to Referee #1**

General Comments This paper examines the future increase/decrease of SSWs due to anthropogenic forcing in a large ensemble of models (12) from the CCMI database. Previous studies have had opposing views on this with some finding an increase in the number of SSWs whilst others have found a decrease. To check the robustness, they use three different commonly-used SSW criteria and conclude that they do not find a significant increase in the future number of SSWs over the course of the 21st century. I find the results to be interesting in that, by using a much larger number of models than previously used, the overall number of SSWs will remain approximately the same. The paper is well-written and the topic is well discussed, hence I suggest only minor corrections which I list below.

Thanks a lot for your comments. Here is our reply to the specific and technical comments in black:

Specific Comments

1) Lines51-52; not just due to anomalously large injections of wave activity from the troposphere, but can also be due to the resonance of wave activity such as that first described in Plumb (1981) and shown by Esler and Scott (2005). I would either cite this second mechanism, or just change the wording in your current sentence to be less definite.
We have modified the sentence to be less definite.

2) Line 53-54; Small point, but not all SSWs can impact the troposphere (e.g., Gerber et al. 2010, GRL and Hitchcock and Simpson 2014). I would instead just add the word 'can' into the sentence to change it from meaning all as it is currently: 'SSWs can also impact the. . .'
Done

3) Line 151; which criteria are absolute and which are relative? Can you somewhere distinguish between the two – from what I can see, the U6090N and WMO are absolute and ZPOL is relative.
It is stated at the end of Section 2.2 (old Line 136) which criteria are absolute and which one is relative. Nevertheless, we have also included a reminder in Section 3.1 (Line 163 in the marked-up manuscript).

4) Line 184-185; at which confidence level do the models simulate statistically significant differences between the SSW duration in the past and in the future? Can you make a comment here on the 90% level? Perhaps just say how many models would become significant if this level was used. My guess is that the HADGEM3-ES and MRIESM-1r1 would be significant at close-to the 95% level as the error bars are nearly separated.
In the case of the multimodel mean, the future change in the duration of SSWs is only statistically significant at the 80% confidence level. When examining each individual model, only HadGEM3-ES shows a statistically significant change in the SSW duration at the 90% confidence level. Apart from HadGEM3-ES, the referee also suggests that the future change for MRI-ESM1r1 might be also statistically significant at the 90% confidence level. However, in that case, this difference would be statistically significant at the 71% confidence level. Probably, the referee got that impression because the error bar of the past extends up to the mean value of the future period and so, it is not easy to identify the end of the first bar.
Following referee's suggestion, we have made a comment on the model that simulates statistically significant differences in the SSW duration between the past and the future at the 90% confidence level (L199-200 in the marked-up manuscript).

5) Line 191; the eddy heat flux (v'T') is what you plot right? This is a proxy for the injection of wave activity. Can you make this clearer? Further, I gather from figure 3 that you use 100hPa (I think this should be included here in the text also), but given the recent paper by de la Camara et

al. (2017) and Birner and Albers (2017), who suggest that 100hPa is not an ideal surface to use as it is already in the bottom of the vortex, how sensitive are you results to the level of choice? Do you get more significant results if you use a slightly lower level?

The referee is correct. Figure 3 shows anomalous eddy heat flux at 100hPa and averaged between 45º-75ºN (aHF100). We have described that in Section 2.3 and included a small comment in new Section 3.4 to make it clearer (L217-218 in the marked-up manuscript).

We acknowledge that these 2 recent papers show that 300hPa is maybe a better level to use than 100hPa. However, 100hPa is the traditional metric and our choice is in line with all previous work. More importantly, we do not have output at 300hPa for some models and so, we cannot do these calculations.

6) Section 2.2 and Line 197; I would be interested to see what the changes between the number of splits and displacements are between the past and the future. From figure 3 it appears that the wave 1 and wave 2 forcing through 100hPa doesn't change too much between each period and so the number of splits and displacements may not change too much either. But given the relatively short length of the paper as it is, this would be an interesting addition.

Thanks for the suggestion. We had indeed started looking at the number of split and displacements SSWs in the past by applying a similar algorithm to Charlton and Polvani (2007). However, the results showed a bias of most models towards an unrealistically high number of displacements events, probably due to a too strong climatological wavenumber 1 wave component in December and January. Thus, given that models could not realistically reproduce the distribution of split and displacements SSWs in the past, we decided not to explore this further. However, figure 3 of the manuscript suggests a null change in the number of splits and displacements as the referee indicates. In the revised version and based on referee's suggestion, we have included a short comment about this when describing figure 3 (L224-225 in the marked-up manuscript).

Technical Comments/Grammatical Errors

1) Line 56; on → at
Done.

2) Line 82; The recent paper by Kim et al. (2017) which you cite later may be a good citation here.
Included!

3) Line 111; Could you clarify this sentence to say whether both reanalyses extend back to 1979, or that both extend back to before 1979 (and if the latter, then which year: 1960?).
They extend back to before 1979, in particular JRA-55 data starts in January 1958 and the ERA-40 reanalysis data is available since September 1957. We have included this information in the manuscript to clarify the mentioned sentence.

4) Line 125; Is the Polar Cap area weighted? I think it should be and it would be good to include a sentence here saying so.
Yes, the polar cap is area-weighted. We have included a comment about that in L134 in the marked-up manuscript.

5) Line 191; 'in the course' → 'during the course'. Also, aHF100 in the text should correspond to the figure title of HF100.
We have modified both things.

**6) Line 193; None of the individual models show significantly different results?**

We have only found some slight significant changes in two models (GEOS-CCM and EMAC-L47), but only for a few days preceding the SSWs. Thus, we think that it was not worth important to report in the paper.

**7) Line 209; 'no statistically significant changes'**

Modified!

**8) Line 220; Can you give references to the figure which shows this? Figure 1?**

Yes, it is Figure 1. We have included the reference to that figure.

**9) Line 221; 'across' → 'using'**

We prefer using 'across' instead 'using' as the slight future increase in frequency of SSWs is detected in all cases when applying each criterion separately.

**10) Line 260; 'in the last years' → 'in recent years'**

Changed

**11) Line 261; 'metrics' → 'metric'**

Changed

**References**

Charlton, A. J., and Polvani, L. M.: A new look at stratospheric sudden warmings. Part I: Climatology and modelling benchmarks, J. Climate, 20, 449-469, 2007.

---

## Author Comment (AC2) · 29 Jun 2018

**Reply to Referee #2**

The paper is easy to read and the main message and methodology are clear. In general I feel positive about the paper and the conclusion but I do think the paper lacks much in-depth analysis above what has been done in previous work, most notably Kim et al. 2017. Some of the figures in the supplementary material might be worth including in the main manuscript (Fig S3 for example). I suggest a major revision to address the points below.

Thanks a lot for your comments. We have addressed all referee's comments. In particular, we have highlighted more clearly how our results compare and extend upon other previous studies, particularly Kim et al. (2017). We have kept the figures in the supplementary material because they are devoted to the analysis of individual models, and these models agree in the main results. Thus, we think showing the multimodel mean values and its robustness across models for some variables is enough and highlights more easily the main conclusions. Nevertheless, we have extended the work and included new analyses such as the deceleration of the polar night jet during SSWs.

Here is our response to your general, specific and technical comments:

General Comments

1) While this study looks at the CCMi models, which is a likely improvement in terms of interactive chemistry and stratospheric processes, it would be nice if this study more clearly explained how its analysis improves or expands upon those of Kim et al. 2017, which considered a large number of stratosphere-resolving (and non-stratosphere resolving) CMIP5 models and two different SSW definitions (and the more extreme RCP8.5 scenario rather than the RCP6.0 scenario used here). From what I can tell, the results were very similar in both studies (an increase in SSWs in future climate scenarios, though not a significant increase), though the message is quite different. While emphasizing the non-significance of the trend does seem important, the results are basically the same. It would be nice if this analysis had included some more in-depth analysis of the CCMi models in particular, maybe of whether models that had different characteristics (prescribed SSTs vs coupled, internal QBO vs nudged, solar variability or no) had different changes in SSWs. The two points below also outline some areas where more analysis could be considered.

First of all, we would like to highlight that the focus of our study and Kim et al's is different. Whereas ours focuses on the effects of projected climate change on the occurrence of SSWs in CCMI runs, the main goal of Kim et al is to search for a new definition of SSWs that is not sensitive to possible model biases. Kim et al perform indeed an in-depth analysis of the new algorithm in the present period in reanalysis data and CMIP5 models. As a secondary task, they apply the new algorithm to RCP8.5 simulations mainly to examine at what extent the application of their new algorithm may affect the conclusions for future frequency of SSWs. In contrast, in our study, we are interested in the future changes in SSWs and so, we do not only investigate the future changes in the mean frequency of occurrence of SSWs by applying different criteria, but we also examine other aspects such as duration of events, deceleration of the polar night jet or the preceding wave activity. Indeed, as far as we know, this is the first study that also compares the sensitivity of these other features to climate change in several models. In all these SSW features, not only the mean frequency, we do not find a statistically

significant future change, and so, that would lead us to highlight the null future change. In the revised manuscript, we have indicated more in detail the strengths of our study in the Introduction and the discussion section (L97-100, and L241-243 and L272-275 in the marked-up manuscript, respectively).

Regarding the suggested analysis of the CCMI models, we agree that it would be a nice exercise to further compare the results for models with different characteristics, if we had found a variety of results among models. However, the same conclusion is reached for all CCMI models, a null change of SSWs in the future. It is true that in the case of the mean frequency of SSWs, a few models show a statistically significant change for one or two criteria. However, the number of these models is so low that we cannot derive any conclusions. Thus, it seems that the different characteristics of the models do not play a relevant role in the impact of climate change on SSWs. We thank the reviewer for this suggestion because it has helped us to highlight this extra point concerning the null change of SSWs. Thus, we have added a short comment about that in section 3.1 (L182-184 in the marked-up manuscript) and another bullet to our main conclusions in Section 4 ("The absence of a future change in SSWs is a robust result across all models examined here, regardless of their biases or different representation of the QBO, coupling to the ocean, solar variability, etc..").

2) This study does look at common SSW definitions, but given that the changes in mean zonal winds at 60N in Figure 4 are barely significant, it would be nice to consider zonal wind reversals at a broader range of latitudes. The authors did look at the 60-90N averaged zonal winds, but if that's cosine weighted it will be dominated by winds near 60N. 60N also seems right on the node between significant weakening winds and significant strengthening winds. It would be interesting to explore (and perhaps more clearly quantify) whether those models that showed no increases (or reductions) in SSWs had significant strengthening mid-latitude winds extending further north (seems to be true considering Fig S3). This might beg the question then, if the jet itself is shifting in the future, is maintaining a definition fixed at a particular latitude like 60N the best way to detect changes? I do agree that the polar cap anomaly/U60-90N results suggest it might not matter too much where it's defined, but exploring that sensitivity more methodically might be useful.

Thanks for the suggestion! We have explored the sensitivity of the results to the latitude chosen for the reversal of the wind in the definition of SSWs (Fig. R2.1). To do so we have computed the mean frequency of SSWs based on the WMO but imposing the reversal of the wind at 55°, 65° and 70°N. First of all, we can see that the main conclusion of our study does not change and most of the models do not show a change. It is true that in a few cases, if we change the latitude of reference, the future change becomes statistically significant at a latitude point. However, only one model (EMAC-L47) shows a systematic significant increase in the SSWs frequency for reference latitudes higher than 60°N. EMAC-L47 is one of the models that displays a latitudinal dipole in the future changes of the climatological zonal winds, as suggested by the reviewer, but other models show that dipole (e.g.: SOCOL3 or IPSL) and there is not a systematic change in the significance of results for higher latitudes.

Although performing this extra analysis was a good suggestion, the results are not relevant for our conclusions and we decided not to include it in the revised version. We prefer to keep the main message of our study clear and avoid confusion in the reader.

[Figure]

**Figure R2.1.** Mean frequency of sudden stratospheric warmings per decade for the past (blue line) and the future (orange line) in function of the latitude selected for the reversal of the wind. Black squares indicate that future vales are statistically significantly different from the past ones at the 95% confidence level.

3) While it does seem clear that there is no significant change in the frequency of SSWs in this analysis, the fact that there appears to be a quite robust weakening of the polar vortex (in a mean sense) is only briefly pointed out in the Discussion. It's worth keeping in mind that since the SSWs represent the tail end of the zonal wind distribution, they may be much more sensitive than the mean to small sample size and higher order moments (e.g., changes in skewness). The authors do mention that the "broadness" (variability) of the distribution does not seem to be changing over time- is it possible that the distribution becomes more skewed? I wonder though whether this weakening of the mean state could have potential climate impacts even if the most extreme events (SSWs) are not significantly changing.

As the referee indicates, the variability of the zonal mean zonal wind at 10hPa had been previously examined and compared in the past and future in each model and we did not find statistically significant differences. Following reviewer's suggestion, we have also plotted the pdf of the zonal mean zonal wind at 10hPa and 60°N for the two periods in typical winter months (December, January, and February) to investigate further possible changes in the distribution of this variable (Fig. R2.2). However, we do not find a future change in the probability distribution of the wind in models. Only a slight shift of the mean value in the future towards lower values is detected in a few models (NIWA-UKCA, CCSRNIES3.2, SOCOL3 and EMAC-L90). This result supports our statement of Section 4 about the lack of statistical significance of the weakening of the vortex in most individual models. The shape of the distribution does not change either, particularly when referring to its asymmetry. The skewness is negative in both periods of study, reflecting that westerly winds tend to be of large amplitude in winter, but the asymmetry is not large though, as the values are not smaller than -0.5 (see values in text boxes of Fig. R2.2).

Based on this additional analysis, we have added a short comment about the general null change of the distribution of the wind data in the past and future period in Section 4. The new statement complements the previous sentence where we had already commented that the future

weakening of the vortex was not statistically significant in most of the models (L266 in the marked-up manuscript). We have included the figure in the Supplementary material as Fig. S4.

[Figure]

**Figure R2.2.** Probability distribution function of the daily zonal mean zonal wind at 60ºN and 10hPa in December, January and February in the past (blue line) and future (orange line) for each CCMI model. The skewness of the distribution in each time period is indicated in the text boxes

**Specific Comments**

Line 63-64: Some of the more recent papers on this topic should be cited here, including Kim et al. 2017 and Manzini et al. 2014, rather than only at the end.

We have included Kim et al. 2017, but not Manzini et al. 2014 because the latter does not examine the future changes in sudden stratospheric warmings.

Line 85: The Kim et al. 2017 results should also perhaps be mentioned here, because they investigated the definition sensitivity as well

Included!

Line 105-107: Would it be possible to consider 40-year blocks from 1960-2100 either by moving the center of the 40 years by ~5-10 years over the full period and getting a distribution that way, or by using a smaller time period (20-30 years) and looking at the change in frequency of consecutive 30-year periods over the entire run? I wonder whether that would give you a better sense for how variable the SSW frequency can be for any given 40 year period (maybe the variability between periods is much larger than the trend between the first and last period).

Thanks for the suggestion! Given that SSWs occur randomly and not very often per decade, we found it difficult to get a distribution just picking-up 40-year block by moving the center of 40

year by ~5-10 years. However, inspired by reviewer's suggestion, we have plotted the evolution of these 40-yr blocks of SSWs computed every 10 years (Fig. R2.3). This procedure allows us to visualize whether there is indeed a trend in the occurrence of SSWs or if in contrast, the multi-decadal variability of SSWs is comparable to the change between the past and future period. Only in a few models, and more specifically in EMAC-L90, do SSWs show a clear increasing trend in the future under climate change conditions. The result agrees well with the statistical analysis performed when just comparing the frequency of SSWs in the past and future periods and applying a Student t-test (Figure 1 of the manuscript). Given the agreement in the results, Figure R2.3 adds little to the study; therefore we have not included it in the revised version of the manuscript

[Figure]

**Figure R2.3.** 40-yr running-mean frequency of SSWs per decade in each CCMI model.

Line 119-121: How did you deal with the temperature criteria here; was the zonal wind first detected and then the temperature gradient had to reverse within a certain number of days?

We have imposed the simultaneous occurrence of zonal mean easterly winds at 60°N and 10hPa and a positive difference of zonal mean temperature at the same level between the pole and 60°N. We have modified the sentence to clarify it.

Line 137-140: Could any other metrics be considered in addition? These are both interesting features but other metrics like the amplitude or depth of the reversal could also be worth considering (to try to further quantify whether these events will still produce significant surface impacts in the future).

As part of a good characterization of SSWs, we have computed the associated deceleration of the polar night jet at 10hPa and the results are included in the new Section 3.3 and Figure 2b. Similarly to other SSW features, the MM value does not show a statistically significant change at the 95% confidence level and only two out of 12 models do show a reduction. Thus, the results support the null future change of SSWs.

Regarding the possible impacts of SSWs on surface, it is important to remark that there is still not a clear knowledge about the factors that can modulate the amplitude of this signal in the troposphere. The internal tropospheric variability is much larger than the stratospheric contribution and so, often masks the stratospheric fingerprint in the troposphere (Gerber et al., 2009). For instance, specifically looking at the amplitude of the stratospheric anomalies, Runde et al. (2016) found that strong stratospheric perturbations do not obligatory have associated a strong surface signal.

Technical Corrections

1) Line 56: change to "weather forecasts on intraseasonal timescales"

Done!

2) Line 111: ERA-40 and JRA-55 extend further than 1979, is that what you mean? Maybe instead of "back of", change to "beyond"?

Yes, that is what we mean. We have modified the sentence and clarified the paragraph to make it clearer.

3) Line 208: change to "possibly accounts for at least some of the mismatch between"

Changed!

**References**

Gerber, E. P., Orbe, C., and Polvani, L. M.: Stratospheric influence on the tropospheric circulation revealed by idealized ensemble forecasts, Geophys. Res. Lett., 36, L24801, doi: 10.1029/2009GL04091, 2009.

Runde, T., Dameris, M.,Garny, H., and Kinnison, D. E.: Classification of stratospheric extreme events according to their downward propagation to the troposphere, Geophys. Res. Lett., 43, , 6665–6672, doi:10.1002/2016GL069569, 2016

---

## Author Comment (AC4) · 29 Jun 2018

**Supplementary material of the manuscript "No Robust Evidence of Future Changes in Major Stratospheric Sudden Warmings: A Multi-model Assessment from CCMI" by B. Ayarzagüena et al.**

5

[Figure]

**Figure S1.** Mean frequency of stratospheric sudden warmings per decade identified with the WMO (green), u6090N (purple) and ZPOL (yellow) criteria in all models, the multimodel mean (MM), JRA-55 and ERA-40 in the past period.

10

[Figure]

**Figure S2. Climatology of 5-day running mean of zonal mean zonal wind at 10 hPa in the past period (1960/61-1999/2000). Shading interval: 5 m s⁻¹.**

[Figure]

**Figure S3.** Future-minus-past differences in the climatology of 5-day running mean of zonal mean zonal wind at 10hPa. Shading interval: 2 m s$^{-1}$. Dots indicate statistically significant differences at a 95% confidence level.

5

[Figure]

**Figure S4. Probability distribution function of the daily zonal mean zonal wind at 60ºN and 10hPa in December, January and February in the past (blue line) and future (orange line) for each CCMI model. The skewness of the distribution in each time period is indicated in the text boxes**

5